# The impact of pediatric early warning score and rapid response algorithm training and implementation on interprofessional collaboration in a resource-limited setting

Samantha L. Rosman[1]☯*, Christine Daneau Briscoe[2]☯*, Samuel Rutare[3], Natalie McCall[4], Michael C. Monuteaux[1], Juliette Unyuzumutima[3], Agnes Uwamaliya[3], Janvier Hitayezu[3]

**1** Division of Pediatric Emergency Medicine, Boston Children's Hospital, Boston, MA, United States of America, **2** Division of Hematology, Boston Children's Hospital, Boston, MA, United States of America, **3** Department of Pediatrics, Centre Hospitalier Universitaire de Kigali (CHUK), Kigali, Rwanda, **4** Department of Pediatrics, Yale University School of Medicine, New Haven, CT, United States of America

☯ These authors contributed equally to this work.
* Samantha.rosman@childrens.harvard.edu (SLR); christine.briscoe06@gmail.com (CDB)

## Abstract

### Introduction

Improved teamwork and communication have been associated with improved quality of care. Early Warning Scores (EWS) and rapid response algorithms are a way of identifying deteriorating patients and providing a common framework for communication and response between physicians and nurses. The impact of EWS implementation on interprofessional collaboration (IPC) has been minimally studied, especially in resource-limited settings.

### Methods

The study took place in the Pediatric Department of the main academic referral hospital in Rwanda between April 2019 and January 2020. Pediatric nurses and residents were trained on the use of the Pediatric Warning Score for Resource-Limited Settings (PEWS-RL) and a rapid response algorithm. Training included vital sign collection, PEWS-RL calculation, IPC and rapid response algorithm implementation. Prior to training, participants completed surveys on IPC with Likert scale responses (from "strongly disagree" to "strongly agree"). Follow-up surveys were then administered nine months later and also included an open-response question on the impact of the PEWS-RL implementation on IPC.

### Results

Sixty-five (96%) nurses were trained and completed the pre-survey and thirty-seven (54%) of the trained nurses completed the post-survey. Twenty-two (59%) pediatric residents were trained in the workshop and completed the pre-survey and twenty-four physicians (4 pediatricians (40%) and 20 pediatric residents (53%)) completed the post-implementation survey. There was a statistically significant increase in the percent of nurses indicating strong

**Data Availability Statement:** All data files are available from the Harvard Dataverse. https://doi.org/10.7910/DVN/EBSOLC.

**Funding:** This study was funded by a grant from the Boston Children's Hospital Global Health Program awarded to SLR. No grant number was assigned. https://www.childrenshospital.org/programs/global-health The funder had no role in study design, data collection and analysis, decision to publish, or preparation of the manuscript.

**Competing interests:** The authors have read the journal's policy and the authors of this manuscript have the following competing interests: This study was funded by a grant from the Boston Children's Hospital Global Health Program. The funder had no role in study design, data collection and analysis, decision to publish, or preparation of the manuscript.

agreement across all domains of communication and collaboration from the pre- to the post-survey. Although the percent of physicians indicating strong agreement increased in the post-survey for all items, only the "share information" item was statistically significant.

## Conclusion

Training and implementation of a PEWS-RL and a rapid response algorithm at a tertiary hospital in Rwanda resulted in significant improvement of nurse and physician ratings of IPC nine months later.

## Introduction

Interprofessional collaboration (IPC) is a partnership between members of the healthcare team that enables a coordinated approach to making healthcare decisions [1,2]. IPC requires regular communication and interaction between members of the healthcare team that respects the contributions and perspectives that each member brings to the care of the patient.

Pediatric inpatient mortality rates in resource-limited settings remain unacceptably high. We know that ineffective communication among health care professionals is one of the leading causes of medical errors and patient harm [3]. Lack of communication and power imbalance can significantly affect coordination of care and patient outcomes. Strong IPC can improve healthcare quality, decrease patient complications, hospital admissions, length of hospital stay, mortality rates, and result in improved patient outcomes [3–5]. Improved IPC can also lead to decreased staff turnover and decreased tension and conflict among caregivers. Although IPC plays an essential role in healthcare quality and outcomes, physicians and nurses seldom receive training on interprofessional collaboration or participate in interprofessional education or training sessions [5]. Resource-limited settings may be especially at-risk of severe consequences due to breakdowns in IPC given insufficient staffing ratios, pronounced hierarchies, and significant differences in backgrounds, training levels and frames of reference between different disciplines within the healthcare team [6].

Early Warning Scores (EWS) are tools designed to detect the early deterioration of inpatients with the goal of early intervention and thereby reduced inpatient morbidity and mortality. They are often used in conjunction with a rapid response team (RRT) or medical emergency team (MET), a team of healthcare professionals tasked with responding to patients at the first signs of deterioration and implementing emergency treatment or transfer to the ICU. In resource-limited settings without available rapid response teams, escalation algorithms have also been used to guide increased frequency of monitoring or consultation with physician teams [7]. The use of EWS in combination with a RRT or escalation algorithm has been found to be associated with fewer clinical deterioration events and emergency resuscitations [8,9]. The few studies of rapid response teams or escalation algorithms in resource-limited settings have also shown promise in reducing deterioration events and ICU transfers [7,10].

EWS systems allow input from both nurses and physicians by using a validated tool for identification of and response to clinically deteriorating patients [2,11]. EWS can empower nurses by providing tools and policies with which to overcome hierarchical or sociocultural barriers to communication and can provide a common reference point and language across the healthcare team [11,12].

Empowering nurses with the knowledge, skills and confidence to be active members of an interprofessional healthcare team has been shown to result in an improved culture of patient

safety and improved patient outcomes [13]. Furthermore, nurse empowerment has been shown to result in decreased burnout [14,15], improved physical and mental health [14], decreased turnover [16], and improved job satisfaction [17].

In a prior study we described the development and validation of a novel Pediatric Early Warning Score for use in Resource-Limited settings (PEWS-RL) at a tertiary referral hospital in Kigali, Rwanda [18]. This tool had been incorporated into the patient files since 2016, but no rapid response system had yet been implemented and there had been little training on PEWS calculation. There were printed instructions written by hospital management at the bottom of the scoring sheet that instructed the nurse to notify the physician immediately if the initial score was 3 or greater or if the score increased by 3 points or more in 24 hours. However, there had been no validation of those scoring cut-offs, no training done around these instructions and no algorithm in place for who to contact or how the physician contacted should respond. Both nursing and physician leaders within pediatrics indicated that few, if any, elevated PEWS scores led to physician notification. In this study we assessed the impact of PEWS-RL and rapid response algorithm training and implementation on interprofessional collaboration and communication within the pediatric healthcare team at a tertiary hospital in Kigali, Rwanda.

## Methods

### Setting and participants

This study took place at the Centre Hospitalier Universitaire de Kigali (CHUK) in Kigali, Rwanda, from April 2019 to January 2020. CHUK is a large tertiary academic referral hospital in Kigali, Rwanda, which receives approximately 70% of the referred cases from hospitals across the country [19]. The Pediatric Department of CHUK consists of 84 beds and is divided into four units: Pediatric Emergency Department (PED) (9 beds), Pediatric Wards (56 beds including 4 PICU beds), Pediatric Outpatient Department (OPD), and Neonatology (20 beds). In 2018, the Pediatric Department admitted 3521 patients with a mortality rate of 7.5%. Of the total Pediatric Department admissions, 159 were admitted to the PICU [20].

At the time of the study, the Pediatric Department included 69 nurses, 10 pediatricians and 38 pediatric residents (approximately a third of whom are working at CHUK at any one time as part of their rotation at four different teaching hospitals). Of the pediatric nurses at the time of the study, 48 had an advanced diploma (A1—completed 3 years of post-secondary education), 15 had a bachelor's degree (A0—completed 4 years of post-secondary education), and 6 had a master's degree in nursing.

This study was approved by both the Boston Children's Hospital Institutional Review Board (IRB-P00030723) and the CHUK Ethics Committee (EC/CHUK/02/2019). Written consent for participation was waived based on the fact that survey responses were fully anonymized.

### Training

Pediatric nurses and residents participated in a one-day workshop on the implementation of PEWS-RL and a rapid response algorithm. The morning sessions were solely for nurses in order to allow time to practice vital signs assessment and PEWS calculation and then the afternoons brought together the pediatric residents and nurses in a single training session to create a collaborative, interprofessional learning environment. While attending pediatricians were invited to these training sessions, none were able to attend so a separate abbreviated training session was conducted during a faculty meeting for attending pediatricians.

We conducted four workshops over the course of two weeks in order to capture as many nurses and physicians as possible. CHUK pediatric nurse and physician leaders lead the workshops in a combination of English, French and Kinyarwanda. Training was delivered using interactive presentations, group discussions and simulation sessions.

During the morning sessions nurses practiced assessing vital signs, respiratory distress and mental status, and using this information to calculate the PEWS-RL score. This was done with a mix of simulated patients (actors) on which they performed live vital signs assessments, videos of patients from which they had to determine their clinical assessment and then were given vital signs, and mannequin-based simulations in which vital signs were provided and they were given a verbal description of their behavior and respiratory exam.

Teams of nurses rotated through five simulation scenarios to practice using the PEWS-RL and rapid response algorithm. After calculating the PEWS-RL score, nurses practiced communicating their concern and then escalating their concerns utilizing the rapid response algorithm. Simulation scenarios were generated by our team including local nursing leadership and local pediatricians in the department based on the most common reasons nurses and residents gave for residents being unable to respond to calls indicating clinical concerns. In these simulations, nurses faced a resident stating they were too busy in the emergency department to respond at that time, a resident who refused to come because he was in a lecture who then did not show up within the expected time for response, a situation in which the resident assigned to the ward was post-call and the covering resident did not respond to their calls, and a case in which the covering resident was not responding and they had to escalate their concerns to the PICU resident (Fig 1).

During the physician and nurse combined teaching afternoons, physician/nursing teams rotated through joint simulation scenarios again using a combination of actors, videos and

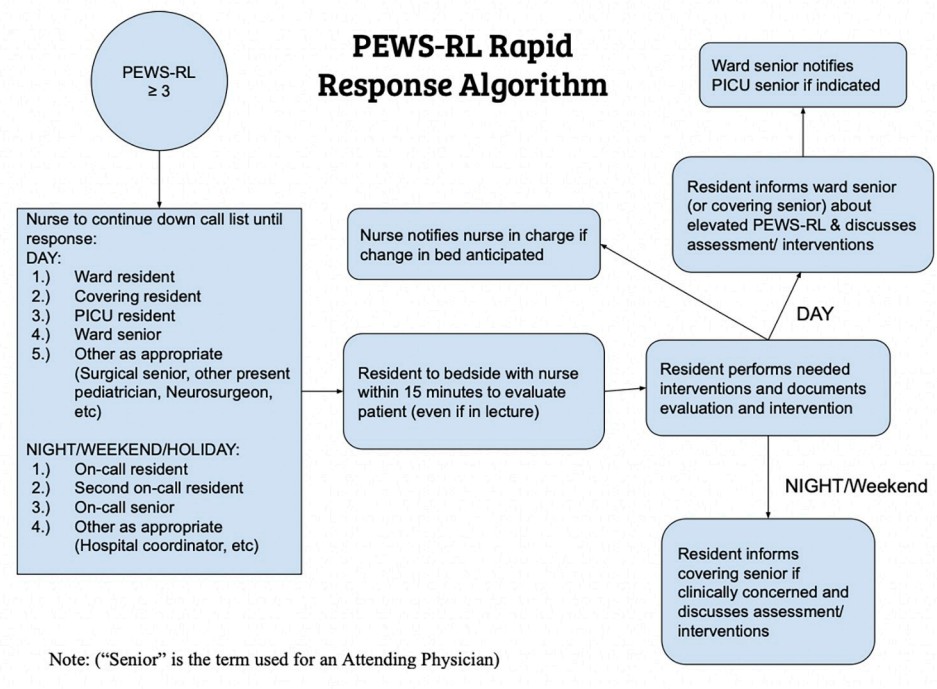

**Fig 1. CHUK PEWS-RL rapid response algorithm.**

mannequins. Simulation scenarios were again generated by our team including local pediatricians and pediatric nursing leadership. This time, instead of the facilitator playing the role of the residents, the residents themselves gave a scripted response to the nurses' calls. These responses included residents who said they could not respond due to other emergencies or teaching conference, residents who were post-call or not responding, forcing escalation to second and third call providers on the algorithm, and a resident who responded that they had assessed the patient that morning and did not think they needed to come back to reassess the patient. Nurses worked through the steps of conveying their concern for serious illness, reiterating the elevated PEWS-RL score, informing the resident of the requirement of bedside assessment in the algorithm, and offering to call the next person in algorithm if the resident was unable to come. Once the urgency of the evaluation was adequately conveyed, the simulation progressed to the resident responding to the bedside. On arrival of the resident "in person" to the bedside of the simulated case, they performed a patient assessment and simulated initial clinical interventions such as dextrose or fluid resuscitation, medication administration, respiratory support or further laboratory or imaging studies. Simulation scenarios were each fifteen minutes followed by five minutes of debriefing.

## Data collection

Pre- and post-workshop survey questions were based on prior studies of IPC assessment and then questions were developed by our research team consisting of Rwandan pediatricians and pediatric nurse leaders and U.S. pediatricians and a pediatric nurse (all of whom had worked and lived in Rwanda for over a year (ranging from 1–10 years)) [21,22]. Questions were tested for clarity and content validity with a small group of local nurses who provided feedback on several iterations of survey items with questions adjusted accordingly. In order to minimize language barriers, surveys were provided in both English and French simultaneously to all participants with translations independently verified by two fluent bilingual French/English-speaking physicians.

On the day of the training, prior to the start of the workshop, the nurses and pediatric residents were asked to complete the anonymous paper-based survey regarding their opinions on the state of IPC and communication. There was one survey for nurses, with questions soliciting their perspectives on how physicians listen to, trust and respond to their concerns, and how physicians collaborate and communicate with them (S1 Survey). There was a second physician-targeted survey focusing on perspectives of physicians on nurse communication regarding sick patients, their impression of accuracy of nurse assessments and level of collaboration between physicians and nurses (S2 Survey). Both nurse and physician leaders felt that the hierarchies in place and expectations around roles of nurses and physicians in this setting demanded different surveys in order to accurately assess views on communication, collaboration and trust. It was felt that a single survey would not invite the honest opinions that we were seeking from nurses on critical issues such as respect for nursing input. Each survey item used a five-point Likert scale ranging from 1 ("strongly disagree") to 5 ("strongly agree").

In follow-up to the workshop, monthly QI data was collated from a very small sample of chart audits (approximately ten per month) and reports and run charts on a variety of process measures around implementation were shared with the nursing leadership to present in the monthly nursing staff meeting. In total, this QI data was presented three times over this nine month period.

While post-testing shortly after training likely would have shown an impact, it was felt that performing an assessment at least 3–6 months after the initial training would better reflect true cultural change in interprofessional collaboration, rather than a transient improvement

following the training. We performed our survey nine months after the workshop to coincide with the completion of the academic year. A link to a nursing version of an online survey (S3 Survey) was emailed to all pediatric nurses who attended the workshops, and a link to a physician version of an online survey (S4 Survey) was emailed to all pediatric residents and pediatric attending physicians regardless of whether they attended the initial training or not (given the baseline lower attendance of this group at the initial training). Survey data were collected and managed using REDCap electronic data capture tools hosted at Boston Children's Hospital. REDCap (Research Electronic Data Capture) is a secure, web-based software platform designed to support data capture for research studies [23,24].

The post-surveys again assessed respondents' opinions on the state of IPC and communication using the same five-point Likert scale, again with one survey version targeted towards nurses and one towards physicians. Respondents were also asked an open-ended question on their opinions on how PEWS-RL implementation changed the way nurses and physicians collaborate and communicate about patient care. Nurses were asked for an open-ended written response to the question "Any additional comments on how you feel the implementation of PEWS has changed the way nurses are respected by physicians when communicating about patient care?" The specific inclusion of the term "respected" was felt by local nursing leadership to be important to invite nurses to discuss this sensitive topic. The hierarchy between nurses and physicians in the local context does not generally permit nurses to openly discuss a lack of respect by their physician colleagues. Therefore, nursing leadership felt that if the question mirrored the physician question asking about communication alone, that nurses would not feel empowered to address the issue of respect. Physicians were asked for an open-ended written response to the question "Any additional comments on how you feel implementation of PEWS has changed the way nurses and physicians communicate regarding patient care?".

While matching pre- to post- surveys within individual respondents would have been the optimal methodological approach, this would have required using a respondent-specific study identifier. Our team believed that, culturally, such an identifier would prevent respondents from trusting the anonymity and therefore from answering questions honestly on such sensitive topics. Instead, we kept surveys completely anonymous to mitigate respondents' fear of any professional repercussions that might result from their responses becoming known to their colleagues.

## Data analysis

**Quantitative data analysis.**   Prior to viewing the Likert scale survey data, we anticipated, based on the culture in this setting, that responses would be skewed towards positive responses, meaning that we would likely need to isolate "strongly agree" from all other responses in order to demonstrate a clinically meaningful change. Based on this expectation, we a priori decided to dichotomize these items as "strongly agree" versus all other responses. We calculated the proportion of "strongly agree" responses for each item, at both the pre- and post-assessments. We also calculated the proportion difference between the pre- and post-assessments, with 95% confidence intervals calculated with a nonparametric bootstrap estimation (with 100 repetitions). Analyses were conducted separately by responder type (i.e., physicians and nurses).

**Qualitative data analysis.**   An inductive content analysis strategy was used to analyze our qualitative responses. Open response answers were recorded directly into a REDCap database. Responses were reviewed to identify positive impacts or barriers to change and each identified phrase was assigned a preliminary label from which a coding scheme was then developed. Data were coded according to the coding scheme using Taguette software (https://app. taguette.org/) with initial coding of English responses by two English-speaking coders and

initial coding of French responses by two bilingual English/French-speaking coders who assigned English codes to the French responses. All coders then subsequently reviewed the responses and codes together as a group and any areas of differential coding were resolved through discussion and consensus. Codes were categorized and then categories of codes were grouped into overall themes. Finally, each comment was again reviewed by all coders to confirm that consensus was reached about the comment belonging to the assigned category and theme.

## Results

### Participants

Sixty-five (96%) of the nurses were trained in the workshops and completed the pre-implementation survey. Thirty-seven (54%) of the nurses who underwent training completed the 9-month follow-up survey. Twenty-two (59%) pediatric residents were trained in the workshop and completed the physician pre-implementation survey. No attending pediatricians completed the training workshop, though a brief training was conducted during a staff meeting. Twenty-four physicians completed the physician 9-month follow-up survey: 4 (40%) attending pediatricians and 20 (53%) pediatric residents.

### Quantitative results

In quantitative analyses, physicians' report of "strong agreement" with survey items in the pre-period ranged from 4.6% ("Nurses are **accurate in their assessment** of patient status") to 72.7% ("When a nurse calls me regarding a patient they are worried about **I always go and assess that patient**"). The proportion reporting "strong agreement" increased in the post-period for all items, but a statistically significant increase was only detected for the item "Physicians **share all information** with the nurses when making decisions on patient care" (Table 1) with an increase of 29.3%.

Among nurses, strong agreement with survey items in the pre-period ranged from 14.0% ("**Decision-making** responsibilities for patients are **shared** among nurses and physicians") to 43.1% ("On my ward physicians and nurses **work together as a team** to monitor and assess patients"). The proportion reporting "strong agreement" significantly increased in the post-period for all items, with proportion increases ranging from 27% ("When I feel there is an **error** made by the physician (verbal or written order) I feel **comfortable notifying** that

**Table 1. Change in survey response, physicians.**

| Survey Item | Pre-Period n = 22 | Post-Period n = 25 | Proportion Difference, 95% Confidence Interval[1] |
|---|---|---|---|
| Physicians **share all information** with the nurses when making decisions on patient care | 5 (22.7) | 13 (52.0) | **29.3 (4.2, 54.3)** |
| **Decision-making** responsibilities for patients are **shared** among nurses and physicians. | 4 (18.2) | 10 (41.7) | 23.5 (-4.7, 51.6) |
| Nurses and physicians **round together** to share patient care information. | 3 (13.6) | 8 (32.0) | 18.4 (-4.8, 41.5) |
| My **opinion is valued** by my colleagues (physicians, charge nurses, matron) when communicating about my patient. | 5 (22.7) | 10 (40.0) | 17.3 (-10.0, 44.6) |
| On my ward physicians and nurses **work together as a team** to care for patients | 6 (27.3) | 9 (36.0) | 8.7 (-17.6, 35.1) |
| Nurses **inform** the physicians in a timely manner **regarding patient deterioration** | 5 (22.7) | 10 (40.0) | 17.3 (-11.3, 45.8) |
| Nurses are **accurate in their assessment** of patient status | 1 (4.6) | 5 (20.8) | 16.3 (-0.1, 33.3) |
| When a nurse calls me regarding a patient they are worried about **I always go and assess that patient** | 16 (72.7) | 20 (80.0) | 7.3 (-17.6, 32.1) |

[1] bolded entries indicate statistical significance.

**Table 2. Change in survey response, nurses.**

| Survey Item | Pre-Period n = 66 | Post-Period n = 37 | Proportion Difference, 95% Confidence Interval[1] |
|---|---|---|---|
| Physicians **share all information** with the nurses when making decisions on patient care | 13 (20.6) | 21 (56.8) | **36.1 (17.2, 55.0)** |
| **Decision-making** responsibilities for patients are **shared** among nurses and physicians | 9 (14) | 19 (51) | **37.5 (18.5, 56.5)** |
| Nurses and physicians **round together** to share patient care information | 19 (28.8) | 24 (64.9) | **36.1 (14.8, 57.4)** |
| My **opinion is valued** by my colleagues (physicians, charge nurses, matron) when communicating about my patient | 22 (33.3) | 22 (62.9) | **29.5 (11.4, 47.6)** |
| On my ward physicians and nurses **work together as a team** to monitor and assess patients | 28 (43.1) | 28 (75.7) | **32.6 (13.5, 51.7)** |
| I feel the **physicians listen and respond** to me when I communicate my concerns regarding patient care | 20 (30.3) | 25 (67.6) | **37.3 (20.7, 53.8)** |
| When I feel there is an **error** made by the physician (verbal or written order) I feel **comfortable notifying** that physician when error is identified | 25 (37.9) | 24 (64.9) | **27.0 (5.9, 48.1)** |

[1] bolded entries indicate statistical significance.

physician when error is identified") to 37.5% (**"Decision-making** responsibilities for patients are **shared** among nurses and physicians") (Table 2).

## Qualitative results

In our qualitative analysis, nurses and physicians commented on positive impacts of PEWS-RL training and implementation in three major categories: 1) Teamwork, 2) Care Improvements, and 3) Respect and empowerment. They identified barriers to improvement in three major categories 1) Not following PEWS-RL/rapid response algorithm protocol, 2) Resource limitations, and 3) Need for more PEWS-RL training.

**Positive impacts.** A large number of both nurses and physicians commented on improved collaboration and communication leading to shared decision-making and better teamwork. Table 3 demonstrates the positive impact categories and themes within those

**Table 3. Positive impacts of PEWS-RL/rapid response algorithm implementation -categories and themes expressed by 37 nurses and 24 physicians.**

| Categories | Themes | Quotations (all *sic*) |
|---|---|---|
| 1. Teamwork | • Collaboration<br>• Communication<br>• Decision-making | "Implementation of PEWS has much changed our collaboration as teamwork as a nurse and physician in our units"–*nurse*<br>". . .we discuss with physicians and we make the decisions together. We work as a team."—*nurse*<br>"Nurses detect early warning signs of the patient and the communication with physicians prevent patients' deterioration. This good communication creates team work and respect"—*nurse*<br>"Doctors and nurses use a same language about the severity of an illness"–*physician*<br>"The implementation of PEWS has significantly changed the way physicians and nurses communicate in regards to patient care, first because it helped nurses, using the score to identify patients who need intervention. The score also indicates when and how they should seek for a physician intervention, which has improved how nurses and physicians communicate."—*physician* |
| 2. Care Improvements | • Early-identification<br>• Fast response<br>• Improved care<br>• Knowledge of vital signs<br>• Prevent deterioration<br>• Reduce mortality | "Implementation of PEWS has helped us to react early in order to reduce mortality"–*nurse*<br>"PEWS is very important because we can identify patient's conditions then decision(s) are taken early to reduce mortality of children."—*nurse*<br>"Really, after having studied PEWS, I have a greater knowledge on vital signs and I testify that I have changed in my decision-making while caring for patients"–*nurse*<br>"We have gained how important is assessment and early intervention. . .and it contributed to the positive outcome in term of patient care"—*physician* |
| 3.Respect/ empowerment | • Respect<br>• Nurse confidence/ proactivity | "After being trained on PEWS the nurses are confident to notify PEWS score because this is evident based on patient's condition"–*nurse*<br>"Nurses are more proactive. Nurses are playing an active role instead of a passive (one)"—*physician* |

**Table 4. Barriers to improvement following PEWS-RL/rapid response algorithm implementation—categories and themes expressed by 37 nurses and 24 physicians.**

| Categories | Themes | Quotations (all *sic*) |
|---|---|---|
| 1. Not following PEWS/ rapid response protocol | • Physicians not following PEWS/rapid response <br> • Nurses not calculating PEWS | "(Doctors) have to take this seriously, because according to me, it is the patients who suffer, or in other words, who are the victims." *-nurse* <br> "Physicians have not respected the implementation of PEWS, there is no change to the physician, they ignore the implementation of PEWS"–*nurse* <br> "For me it remains the same because sometimes is not calculated in the file Nurses used to communicate in case of sick child."—*physician* |
| 2. Resource limitations | • Understaffing <br> • Lack of vital signs monitoring equipment | "Physicians are not enough in number which can affect them to react early"–*nurse* <br> "Avail monitors for taking vital signs as each ward at CHUK has only one monitor it compromises care of patients"—*physician* |
| 3. Need for more PEWS training | | "The new doctors must be informed (of) the PEWS process in the first days of orientation"–*nurse* <br> "More training for nurses and residents as they are primarily (the) one(s) who are with patients everyday"—*physician* |

categories along with selected representative comments. Many commented that PEWS-RL was important to care and resulted in the earlier identification of sick patients, faster response and interventions to signs of worsening illness, the ability to prevent deterioration, and the belief that PEWS-RL reduced morbidity and mortality. One nurse commented on improved knowledge of vital signs. Several nurses expressed that they were more respected by physicians and felt more confident. One physician noticed that nurses were more proactive following PEWS-RL implementation.

**Barriers to improvement.** Some respondents felt that no significant change had taken place and several identified barriers to improvement. The categories of barriers and themes expressed within these categories along with representative quotes are displayed in Table 4. A few nurses expressed the opinion that physician behavior had not changed significantly in response to the PEWS-RL implementation or that physicians were not following the PEWS-RL/rapid response algorithm protocol. One physician commented that nurses were not calculating the PEWS score. Both nurses and a physicians expressed the need for more training on PEWS-RL both for reinforcing skills as well as for training newly rotating physicians.

## Discussion

We were able to demonstrate significant improvements in IPC and communication nine months after training and implementation of the PEWS-RL/rapid response algorithm. Measures of interprofessional collaboration were low on nearly all questions at the start of the study. Nine months later, the post-survey showed significant improvement in all measures by nurses and a trend towards improvement in all measures by physicians with significance reached on the measure assessing information sharing.

The fact that the post-survey was completed nine months after training for both nurses and physicians makes it more likely that answers reflected an enduring change in IPC culture and practices rather than simply a brief behavioral change immediately following the training session. While QI data was shared three times with nurses over that nine month period, no

training reinforcement or refresher sessions were held. Despite a number of physicians not attending the initial training, we still saw an improvement in IPC reported by physicians. We hypothesize that this may be due to an influence of the PEWS-RL/rapid response algorithm system itself on IPC by creating a framework and algorithm to open this channel of communication, rather than simply the training alone resulting in improvements in IPC.

Our PEWS-RL/rapid response algorithm training not only targeted vital signs measurement and PEWS-RL calculation but also created an interprofessional training environment in which to discuss improving communication and collaboration. The training offered both tools to facilitate communication as well as a safe learning environment in which to practice, through the use of simulation, the use of these tools and strategies to overcome communication barriers.

The PEWS-RL/rapid response algorithm itself may have served to remove hierarchical barriers that previously prevented communication while also providing a common language and frame of reference on which to base these communications around clinical deterioration. The collaborative culture, language, and mutually agreed upon triggers and response protocols created by such a system can empower nurses to contact physicians when they see signs of patient deterioration. This EWS/rapid response algorithm can foster a spirit of information sharing and collaborative decision-making.

The qualitative data supports the quantitative results in that the majority of nurses and physicians commented on improved teamwork, improved care, and improved respect and empowerment. It identified barriers to change that must be further explored and integrated into future trainings, resource allocation decisions, and system-based improvements.

Though all survey items were judged to assess important aspects of interprofessional collaboration, some components of IPC may have been more impacted than others by our training and PEWS-RL/rapid response algorithm implementation. Further, some measures may be more closely aligned with the quality of healthcare delivered to patients. Questions were not formulated in such a way as to be able to assess impact on specific domains of IPC but this could be a worthwhile area to explore in future research.

It is hard to speculate about the discrepancy between nurses and physician responses given several confounders and limitations that cannot be measured. The number of respondents was different between the two groups and the survey questions themselves were not identical. While all nurses who completed the post-survey had completed the training, the physician responses included some who completed the training and some who had not. A much higher percentage of nurses completed training than physicians, secondary to scheduling and logistical challenges. Furthermore, given resident rotation schedules, some respondents may have spent the majority of the nine intervening months at CHUK while others may have spent only a brief time there during that nine-month period. Finally, given the subjective nature of responses, they may be significantly influenced by experience, level of education, age and other factors for which we did not control that may differ between nurses and physicians.

Continuous quality improvement is likely a critical step to maintaining an improvement in IPC over time. During this study we provided QI data on PEWS calculation rates, accuracy, physician response rates and other related process measures to nurses and physicians every 1–3 months (depending on our nurse data-collector's ability to collect data from chart audits) that was shared in the weekly nurse meeting or physician staff meeting. Intermittent reviews on calculation of PEWS-RL and the escalation algorithms were done during these nurse and physician meetings.

Rwandan physician and nurses were key leaders of the project for the initial algorithm design, teaching session and simulation scenario development and delivery, as well as for QI measure presentations. We believe local leadership was a critical component of the project's

success. A single training by a visiting team, whose impact is assessed immediately, is quite different than a project that was championed by a local interprofessional team of leaders and repeatedly reinforced.

As demonstrated in the qualitative data, we encountered a variety of barriers to change: resource and staffing limitations, lack of buy-in to the PEWS-RL system and the need for further training. While we were able to overcome some of these barriers, many need further work. Resource limitations were a significant barrier, primarily due to staffing limitations and therefore significant competing demands for both nurses and physicians as well as due to availability of equipment. The tool was developed in such a way as to seamlessly integrate with the existing vital signs collection form previously in the charts. Further, it did not require separate score calculations but rather only required nurses to tabulate check marks in an attempt to limit impact on clinical practice flow and time required for PEWS scoring. While our process measures are soon to be published showing our full data, we had high rates of nurse scoring compliance after our workshops, which nurses attribute to easy integration into their workflow. However, more work is needed as we did not have 100% compliance and suspect that compliance rates will drop with time from training.

Competing demands on resident and attending time were a major barrier that prevented us from universally training all resident and attending physicians. We suspect this lack of physician training contributed substantially to nurses noting that some physicians were not responding to their notifications and not familiar with the PEWS-RL or response algorithm. Lack of physician buy-in was likely due, at least in part, to the lack of participation in training. Without the training, physicians missed an introduction to local and global data on PEWS effectiveness, a discussion of benefits and barriers to implementation, and the chance to practice use of the tool and algorithm. Without an appreciation of the opportunity for IPC and potential resultant benefits to the patients, the rapid response algorithm likely came across as one more demand on their already overstretched time.

The fact that no attending pediatricians were able to attend the training likely had a significant impact on buy-in. Even if residents follow the protocol, if they are met with resistance when they contact their supervising attending, their future compliance with the algorithms and belief in the importance of the tool to patient care will likely erode over time. Similarly, when physicians fail to respond appropriately to nurse notifications, it likely disincentivizes nurses to continue scoring and contacting physicians. While nurses received dedicated time to attend the training covered by a stipend for participation, no such protected time or stipend was provided to physicians. Without dedicated protected time, residents faced competing clinical demands preventing them from attending. Attending pediatricians similarly had competing clinical, administrative, and personal demands including time in private clinic that contributes significantly to their income. Providing clinical and administrative coverage for physicians as well as a stipend so that they are not forced to make a financial sacrifice to attend the training, would likely have improved attendance substantially.

Further study is needed to determine the most effective way to maintain or continue to build on improvements in IPC over time. Continuous IPC improvement may involve periodic re-trainings taught to both physicians and nurses together, periodic assessments of IPC progress and QI reporting of this data, or integration of IPC teaching into regular education sessions. Furthermore, a root cause analysis of morbidity and mortality data or incidents in which elevated PEWS scores were not recognized or to which a response did not occur could help elucidate barriers to effective PEWS-RL/rapid response algorithm implementation or IPC.

This study has several limitations. It was only conducted at a single center (CHUK) where the PEWS-RL had been developed and validated in a previous study and was already familiar

to some of the healthcare team members. It is unknown whether the improved IPC will translate to other hospitals with different patient populations and healthcare workers and less familiarity with, or buy-in to, the PEWS-RL system. If the level of training, staffing, or culture of the setting differ substantially, the PEWS-RL/rapid response algorithm system may have more or less of an effect on IPC. Therefore, replication across a variety of different resource-limited settings is critical to ensuring that similar improvement in IPC measures are achieved. Furthermore, there could have been other concurrent changes effecting IPC during the period of time after implementation of the PEWS-RL/rapid response algorithm, though there is no report by physician or nurse pediatric leadership within the hospital of any related interventions, trainings or systemic changes during this time period. While we did separately assess both process measures and morbidity and mortality before and after training and PEWS-RL implementation (to be published separately), the study was not designed to specifically measure the effect of improved IPC on quality of care. However, given the large number of studies on the impact of communication and collaboration on clinical outcomes it seems reasonable to assume that improving IPC is inherently a good thing for patient care and that further work in this domain could be quite helpful in resource-limited settings.

Our quantitative analysis was limited by our inability to pair data. In our judgement and based on feedback from local nurses, assigning a study ID number would have resulted in significant fear of loss of anonymity and therefore of repercussions based on responses. We believe, therefore, that a lack of perceived anonymity would have substantially biased our survey results. Thus, it was felt preferable to have unpaired data but retain our ability to have survey answers be as honest as possible. Unfortunately, mistrust, hierarchies, and data collection limitations can result in substantial limitations to research methodologies in many settings. Researchers must always balance the need to obtain accurate information with culturally appropriate methods of data collection.

The fact that we were unable to differentiate the post-surveys of the pediatric residents who had attended the initial training (and therefore had completed the pre-survey) from those who had not was a major limitation. It is possible that those who attended the training had significantly different answers than those who had not, so that adding in those who had not attended training to the post-data for physicians could have substantially biased the results for physicians; though we would expect this to dilute any effect size rather than augment it. Fortunately, for nurses, all who completed the post-survey attended the training so there is not a similar potential bias introduced for that group.

Finally, our post-survey response rate only captured 54% of nurses, 40% of attending pediatricians and 53% of pediatric residents. The fact that the follow-up survey was done electronically may have limited response rates as many have limited internet access or have to pay for their own internet data, which can be cost-prohibitive. Unfortunately, our local team did not have the time and resources to distribute and collect paper follow-up surveys across multiple hospitals in which pediatric residents were rotating and to nurses, some of whom had moved on to new assignments, at the planned nine-month post-intervention assessment period. It is possible that those who did not respond had different views than those who did respond, which could bias our results substantially. It is possible this could exaggerate our effect size if those who felt there is a difference in IPC following the intervention were more likely to respond than those who felt there is no difference. Though it is also entirely possible that those who felt the tool was not helpful were more likely to respond in an attempt to advocate against its use, in which case our effect size would be artificially diminished. Given that data was not paired and answers were completely anonymous, we have no way of assessing even baseline characteristics to determine whether there was a significant difference between responders and non-responders.

## Conclusion

In conclusion, this study demonstrates that, in a resource-limited setting, the implementation of PEWS-RL and a rapid response algorithm, with training on clinical skills and interprofessional collaboration, resulted in significant improvement in nurse and physician ratings of IPC nine months later. Providers identified the positive impacts of PEWS-RL/rapid response algorithm implementation being teamwork, care improvements and respect/empowerment. They identified the barriers to improvement as being not following PEWS/rapid response algorithm protocols, resource limitations, and the need for more training on PEWS-RL/rapid response algorithm. Consideration of these barriers is needed during implementation and ongoing training and quality improvement efforts. Further study is needed to assess whether this improved IPC translates directly into improved patient care and reduced morbidity and mortality across a variety of different resource-limited settings. We look forward to sharing our data on improved implementation process measures as well the impact on clinical outcomes in separate publications.

## Supporting information

**S1 Survey. Nursing IPC pre-training survey.**
(DOCX)

**S2 Survey. Physician IPC pre-training survey.**
(DOCX)

**S3 Survey. Nursing IPC post-training/implementation survey.**
(DOCX)

**S4 Survey. Physician IPC post-training/implementation survey.**
(DOCX)

## Acknowledgments

We would like to thank the Pediatrics Department and the CHUK administration for their continuous support of Pediatric Early Warning Scores, quality improvement within pediatrics, and efforts to strengthen interprofessional collaboration. We would like to thank the pediatric nurses, pediatric residents and senior pediatricians for their efforts on this project during and after the workshops were completed. We would like to thank Carole Orchard for her guidance on training and measuring interprofessional collaboration, Trish Milburn for all of her time and support during the workshops and David Mills for his assistance in the training workshops.

## Author Contributions

**Conceptualization:** Samantha L. Rosman, Christine Daneau Briscoe, Janvier Hitayezu.

**Data curation:** Samantha L. Rosman, Christine Daneau Briscoe, Samuel Rutare.

**Formal analysis:** Michael C. Monuteaux.

**Funding acquisition:** Samantha L. Rosman, Christine Daneau Briscoe.

**Investigation:** Samantha L. Rosman, Christine Daneau Briscoe, Samuel Rutare, Natalie McCall, Janvier Hitayezu.

**Methodology:** Samantha L. Rosman, Christine Daneau Briscoe, Natalie McCall, Agnes Uwamaliya, Janvier Hitayezu.

**Project administration:** Samantha L. Rosman, Christine Daneau Briscoe, Natalie McCall, Juliette Unyuzumutima, Agnes Uwamaliya, Janvier Hitayezu.

**Resources:** Juliette Unyuzumutima, Agnes Uwamaliya, Janvier Hitayezu.

**Supervision:** Samantha L. Rosman, Samuel Rutare, Natalie McCall, Juliette Unyuzumutima, Janvier Hitayezu.

**Validation:** Samantha L. Rosman.

**Writing – original draft:** Samantha L. Rosman, Christine Daneau Briscoe, Michael C. Monuteaux.

**Writing – review & editing:** Samantha L. Rosman, Christine Daneau Briscoe, Samuel Rutare, Natalie McCall, Michael C. Monuteaux, Juliette Unyuzumutima, Agnes Uwamaliya, Janvier Hitayezu.

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
