## [Decision Letter · Decision Letter 0]

22 Mar 2022

PONE-D-21-28676The impact of pediatric early warning score and rapid response algorithm training and implementation on interprofessional collaboration in a resource-limited settingPLOS ONE

Dear Dr. Rosman,

Thank you for submitting your manuscript to PLOS ONE. After careful consideration, we feel that it has merit but does not fully meet PLOS ONE’s publication criteria as it currently stands. Therefore, we invite you to submit a revised version of the manuscript that addresses the points raised during the review process.

The manuscript has been evaluated by two reviewers, and their comments are available below.

The reviewers have raised a number of concerns that need attention. They request additional information on methodological aspects of the study, the study discussion, and other minor queries regarding this manuscript. 

Could you please revise the manuscript to carefully address the concerns raised?

We look forward to receiving your revised manuscript.

Kind regards,

Sebastian Shepherd

Associate Editor

PLOS ONE

Reviewers' comments:

Reviewer's Responses to Questions

**Comments to the Author**

1. Is the manuscript technically sound, and do the data support the conclusions?

Reviewer #1: Yes

Reviewer #2: Yes

2. Has the statistical analysis been performed appropriately and rigorously? 

Reviewer #1: Yes

Reviewer #2: Yes

3. Have the authors made all data underlying the findings in their manuscript fully available?

Reviewer #1: Yes

Reviewer #2: Yes

4. Is the manuscript presented in an intelligible fashion and written in standard English?

Reviewer #1: Yes

Reviewer #2: Yes

5. Review Comments to the Author

Reviewer #1: I applaud the authorship team on well written paper and a well thought out study. It is also appreciated that this study builds on prior work at the study site.

Early Warning Systems (EWS) and Rapid Response Teams (RRT) relationships is not so clear to a reader not familiar with these two concepts. The paper could be strengthened if the introduction clarifies this. Have other studies shown a relationship between the two? Also although validated at the study site from prior , it is not clear how it was in use at the site apart from paper charting. You note that PEWS-RL has been validated in a prior study in Rwanda, but is there data showing RRT or rapid response algorithm use in a similar setting?

In the methods section you mention a role-play activity in which nurses had to express the need to escalate care to residents (lines 1611-164). They do seem oriented towards a Western setting. How were these scenarios selected? Was their input from study site staff?

Similarly, more clarity on how the survey tools were developed (lines 183-191) would strengthen the paper. Were these adapted from previously validated surveys or similar studies?

It would be helpful to clarify why the nursing and physician surveys varied.

Authors mention that a follow-up survey was conducted 9 months after the workshop, why 9 months? Were they any refresher workshops in between?

Authors mention “Nine-months after the workshop, an emailed link to an online survey was sent to all nurses in the Pediatric Department who attended the workshops, and all pediatric residents and pediatric attending physicians regardless of whether they attended the initial training or not (given the baseline lower attendance of this group at the initial training).” This seems to be a methodological limitation when comparing pre and post survey responses that was not addressed in the limitations

I am not sure the data presented in the survey (lines 317-320) “Furthermore, the improvement in IPC reported by physicians, despite a number of physician respondents not attending the initial training speaks to the idea that the PEWS-RL/RRT itself is likely contributing to the IPC improvements, rather than the training alone being responsible for this change.” Is really supported by the intervention and the qualitative results

A more robust discussion of how to barriers to change will or were addressed would also strengthen the discussion.

Although the authors mention their large loss to follow-up in the limitations section, the paper would benefit from a more robust inquiry into why this occurred and how it impacted the findings and results.

Reviewer #2: Overall this was a well-conceived study that the authors described very clearly and highlighted relevant data with an illuminating discussion. There are a few modifications that might strengthen the evaluation of this tool and post-training response rate, but authors had good insight into study limitations. It would be wonderful to see this paired with clinical outcomes data in a future study to examine impact of PEWS-RL and RRT training on pediatric outcomes.

6. PLOS authors have the option to publish the peer review history of their article (what does this mean?). If published, this will include your full peer review and any attached files.

Reviewer #1: **Yes: **Catalina Gonzalez Marques

Reviewer #2: **Yes: **Rebecca E Cook, MD, MSc

---

## [Author Response · Author response to Decision Letter 0]

5 May 2022

Many thanks for your careful review of our manuscript "The impact of pediatric early warning score and rapid response algorithm training and implementation on interprofessional collaboration in a resource-limited setting.” We found your suggestions to strengthen the paper very helpful and appreciate the opportunity to submit this revised manuscript. You will find the according tracked changes in the marked-up copy of our manuscript. Please also find the responses to each of the comments in the table below. We find our manuscript to be significantly improve and are grateful for the time, effort and suggestions that have helped strengthen the paper. 

Our financial disclosure is included in our revised cover letter as well as in the document. This study was funded by a grant from the Boston Children’s Hospital Global Health Program awarded to SLR. No grant number assigned. The funder had no role in study design, data collection and analysis, decision to publish, or preparation of the manuscript.

A revised competing interest statement has also been included in the cover letter. The authors have read the journal's policy and the authors of this manuscript have the following competing interests: This study was funded by a grant from the Boston Children’s Hospital Global Health Program. The funder had no role in study design, data collection and analysis, decision to publish, or preparation of the manuscript.

Sincerely,

Samantha Rosman, MD, MPH

Director of Global Health Equity, Boston Children’s Hospital Global Health Program

Pediatric Emergency Physician, Boston Children's Hospital

Instructor, Harvard Medical School

samantha.rosman@childrens.harvard.edu

We have revised our financial disclosure statement and competing interests statement to include all relevant requested information and have included this updated statement in our revised cover letter.

We are resubmitting our figure according to guidelines.

Journal Requirements: 

1. Please ensure that your manuscript meets PLOS ONE's's style requirements, including those for file naming. - We have reviewed the style requirement but have not been able to locate the manuscript naming requirements but have named supporting files and figures according to guidelines.

2. Please include additional information regarding the survey or questionnaire used in the study and ensure that you have provided sufficient details that others could replicate the analyses. For instance, if you developed a questionnaire as part of this study and it is not under a copyright more restrictive than CC-BY, please include a copy, in both the original language and English, as Supporting Information. - We have edited to cite the other IPC survey references from which we drew initial ideas (line 192) and have attached the surveys as supplementary material and referenced them in the relevant section on methods. (line 207, 210, 225, 226)

3. We note that the grant information you provided in th' ‘Funding Informaton’ and ‘FinanciaDisclosure'’ sections do not match. When you resubmit, please ensure that you provide the correct grant numbers for the awards you received for your study in the ‘Funding Information’ section. - We have revised our funding information and financial disclosure statement to include all relevant requested information and apologize for the misunderstanding regarding these instructions.

4. We note that you have stated that you will provide repository information for your data at acceptance. Should your manuscript be accepted for publication, we will hold it until you provide the relevant accession numbers or DOIs necessary to access your data. If you wish to make changes to your Data Availability statement, please describe these changes in your cover letter and we will update your Data Availability statement to reflect the information you provide. - We have uploaded our data to Harvard Dataverse and have included the DOI in our data availability statement.

5. Please review your reference list to ensure that it is complete and correct. If you have cited papers that have been retracted, please include the rationale for doing so in the manuscript text, or remove these references and replace them with relevant current references. Any changes to the reference list should be mentioned in the rebuttal letter that accompanies your revised manuscript. If you need to cite a retracted article, indicate tharticle's’s retracted status in the References list and also include a citation and full reference for the retraction notice. - We have reviewed the reference list and have not cited papers that have been retracted to the best of our knowledge. The only change in the reference list is the addition of the studies from which our survey instrument ideas were initially taken.

Reviewer #1 Comments from “Comments to Authors” Authors’ response

1. Reviewer #1: I applaud the authorship team on well written paper and a well thought out study. It is also appreciated that this study builds on prior work at the study site. - Thank you for your appreciation and your detailed review of our manuscript

2. Early Warning Systems (EWS) and Rapid Response Teams (RRT) relationships is not so clear to a reader not familiar with these two concepts. The paper could be strengthened if the introduction clarifies this. Have other studies shown a relationship between the two? Also although validated at the study site from prior , it is not clear how it was in use at the site apart from paper charting. You note that PEWS-RL has been validated in a prior study in Rwanda, but is there data showing RRT or rapid response algorithm use in a similar setting? - These are great questions that we have answered in the revised version of the manuscript: 

1) We have added information about the relationships between Rapid Response Teams (RRT) and Pediatric Early Warning Scores (PEWS) in the introduction. (lines 75-84)

2) We have clarified the use of PEWS at the study site prior to this study (lines 99-110)

3. In the methods section you mention a role-play activity in which nurses had to express the need to escalate care to residents (lines 1611-164). They do seem oriented towards a Western setting. How were these scenarios selected? Was their input from study site staff? - We clarified (lines 158-161 and 170-171) how the role play scenarios were developed largely by local Rwandan nursing leadership and physician input.

4. Similarly, more clarity on how the survey tools were developed (lines 183-191) would strengthen the paper. Were these adapted from previously validated surveys or similar studies? - We have included reference to the AITCS tool that we reviewed prior to development of our tool. The author of the tool asked that we not state that we adapted her tool but rather just cite it as a reference as she did not want it associated with her if it was not using the exact tool. Therefore, we used ideas and concepts but did not directly utilize her tool as its length and content was not appropriate for our setting. 

5. It would be helpful to clarify why the nursing and physician surveys varied. - This was based on the input received from the local nurses and physician leaders. This is now explained in lines 210-214

6. Authors mention that a follow-up survey was conducted 9 months after the workshop, why 9 months? Were they any refresher workshops in between? - The 9-month interval was chosen based on wanting at least 3-6 months to pass in order to better measure sustained cultural change rather than simply a reflection of recent training, combined with the fact that 9 months coincided with completion of the academic year (for the residents). There was no refresher workshop in between. However, the QI conducted, where a small sample of charts were analysed and process measures presented based on these interval data) could have elicited a discussion of the use of PEWS and the escalation algorithm (see line 216-225)

7. Authors mention “Nine-months after the workshop, an emailed link to an online survey was sent to all nurses in the Pediatric Department who attended the workshops, and all pediatric residents and pediatric attending physicians regardless of whether they attended the initial training or not (given the baseline lower attendance of this group at the initial training).” This seems to be a methodological limitation when comparing pre and post survey responses that was not addressed in the limitations - We agree that this was a methodological limitation and have added to the discussion section to further clarify and discuss potential impacts of this limitation. (line 521-528). 

8. I am not sure the data presented in the survey (lines 317-320) “Furthermore, the improvement in IPC reported by physicians, despite a number of physician respondents not attending the initial training speaks to the idea that the PEWS-RL/RRT itself is likely contributing to the IPC improvements, rather than the training alone being responsible for this change.” Is really supported by the intervention and the qualitative results - We changed the wording to better reflect that this is one of our hypotheses around interpreting the data but that we certainly don’t have an means by which to make a definitive conclusion in this regard. (line 365)

9. A more robust discussion of how to barriers to change will or were addressed would also strengthen the discussion. - We added some of these barriers in the discussion (see lines 431-479) 

10. Although the authors mention their large loss to follow-up in the limitations section, the paper would benefit from a more robust inquiry into why this occurred and how it impacted the findings and results. - These details about the reasons for loss to follow up were added (lines 529-541)

Reviewer # 1 comments (found in comment boxes on the pdf of the manuscript) Authors’ response

In the “Ethical statement” box, Reviewer 1 asked if the correct term was Internal Review Board or Institutional Review Board (for the respective institutions). The reviewer also asked to include the IRB protocol number - We apologize for this mistake and have corrected the terminology appropriately and added corresponding protocol numbers.

Line 155: Reviewer 1 suggested we consider including the PEWS-RL algorithm as this could help provide readers better insight and context as we narrate the scenarios in the training. - This rapid response algorithm was included in Figure 1 and cited in the text but the actual figure was attached at the end of the document as instructed. If there is something we can do to better clarify this or if there was something else intended please do let us know and we are happy to modify as appropriate.

Line 205: Reviewer #1 commented: “ This is a leading question - assumes that PEWS has made nurses feel more respected, is this how question was phrased? would have been better to state the way nurses are treated by physicians when communicating about patient care.” - We recognize that this perhaps could have been worded in a different way but have sought to address the reasons behind this chosen wording in this setting on line 247, where we explain the choice of the word “respected”, which was suggested by the local nursing leadership. They recommended modifying from the question that remains in the physician version which asks about communication, because they felt that secondary to the strong hierarchy between nurses and doctors, failing to call out respect specifically might not open the door for nurses to feel empowered to talk about the sensitive issue of respect. 

Line 220. Reviewer #1 commented: “Was the plan to analyze the scale as dichotomized made a priori in study design or after collecting results?” - Thank you for this question. We have better explained now that this decision was made a priori and why we felt it was necessary. (lines 266-270)

Line 238: Reviewer #1 commented: “What was the congruence of coding?” Our coding discussions were done in a cooperative group in real time so congruence between coders was not separately analyzed but all 4 coders agreed on final codes and categories generated. For clarity, we specified how the data was coded on lines 287-289

Line 322: Reviewer #1 commented: “It would have been interesting to see some of the QI data on how training impacted VS assessment and quality, etc” - While the current paper focuses on the impact of PEWS/rapid response algorithm training on IPC, we did simultaneously collect data on the impact of training on process measures such as VS assessment, PEWS calculations, etc. that will soon be published separately and a third portion of the study on the impact on morbidity and mortality, the publication of which is also forthcoming.

Reviewer #2 Comments Author's response

1. Overall this was a well-conceived study that the authors described very clearly and highlighted relevant data with an illuminating discussion. There are a few modifications that might strengthen the evaluation of this tool and post-training response rate, but authors had good insight into study limitations. It would be wonderful to see this paired with clinical outcomes data in a future study to examine impact of PEWS-RL and RRT training on pediatric outcomes. Thank you for your review and comments. - While the current paper focuses on the impact of PEWS training on IPC we did simultaneously collect data on the impact of training on process measures such as VS assessment, PEWS calculations, etc. that will soon be published separately and a third portion of the study on the impact on morbidity and mortality, the publication of which is also forthcoming. We have added this to our discussion and conclusions to better explain the future directions.

---

## [Editor Report · Decision Letter 1]

8 Jun 2022

The impact of pediatric early warning score and rapid response algorithm training and implementation on interprofessional collaboration in a resource-limited setting

PONE-D-21-28676R1

Dear Dr. Rosman,

We’re pleased to inform you that your manuscript has been judged scientifically suitable for publication and will be formally accepted for publication once it meets all outstanding technical requirements.

Kind regards,

Rebecca Cook, MD, MSc

Guest Editor

PLOS ONE

Additional Editor Comments (optional):

Thank you for your revisions on your paper describing the impact of Pediatric Early Warning Score and Rapid Response Team Algorithm on interprofessional collaboration in Rwanda. Your revisions systematically addressed the suggestions of the reviewers and strengthened the manuscript. This is a well-conceived study that provides relevant data and an insightful discussion and I agree that the manuscript has addressed the suggestions of reviewers. I did find a few small typographical and grammatical errors that should be corrected prior to publication (Attached). I look forward to the follow-up paper focusing on the clinical outcomes data as I think for other sites to consider investing in the implementation of the PEWS and RRT training they would like to see both the interprofessional and the direct patient care impact.
---

## [Editor Report · Acceptance letter]

13 Jun 2022

PONE-D-21-28676R1 

The impact of pediatric early warning score and rapid response algorithm training and implementation on interprofessional collaboration in a resource-limited setting 

Dear Dr. Rosman:

I'm pleased to inform you that your manuscript has been deemed suitable for publication in PLOS ONE. Congratulations! Your manuscript is now with our production department. 

Kind regards, 

on behalf of

Dr. Rebecca Cook 

Guest Editor

PLOS ONE